# Research on the Characteristics of Safety Culture and Obstacle Factors among Residents under the Influence of COVID-19 in China

**DOI:** 10.3390/ijerph20031676

**Published:** 2023-01-17

**Authors:** Qifei Wang, Yihan Zhao, Jian Wang, Haolin Liu, Hui Zhang

**Affiliations:** 1School of Mechanical-Electronic and Vehicle Engineering, Beijing University of Civil Engineering and Architecture, Beijing 102616, China; 2Institute of Urban Systems Engineering, Beijing Academy of Science and Technology, Beijing 100089, China; 3Research Center of Urban Operation Safety, Beijing Academy of Emergency Management Science and Technology, Beijing 101101, China

**Keywords:** COVID-19, safety culture among residents, comprehensive evaluation, obstacle degree model, safety culture questionnaire

## Abstract

This study established a comprehensive evaluation indicator model for the safety culture among residents during COVID-19 and an obstacle degree model for the identification of the major factors affecting the residents’ safety culture. The results show that the overall level of the safety culture among residents was 0.6059. Safety education, channels for learning knowledge regarding safety, and implementation of safety management systems are currently the major obstacles affecting safety culture among residents, but there is still space for improvement in the future. Furthermore, the level of safety culture was strongly related to the distance from the infected, because this changes the risk of viral infection. There are also differences in obstacle factors in different regions. Therefore, it is necessary to implement measures targeting the improvement of safety culture in accordance with the risk of viral infection. Strategies for strengthening the safety culture are also given in this study for consideration in strategic decision making with the aim of promoting the improvement of safety culture among residents, which may help to reduce the risk of infection with COVID-19 for residents.

## 1. Introduction

Since 2019, the global spread of COVID-19 has had a serious impact on people’s lives, resulting in many people becoming infected and dying. Community is the basic unit of life. It is also one of the places with the highest rates of occurrences of clusters of COVID-19 infection. In the early stages of COVID-19, the epidemic raged out of control because of the weak construction of a safety culture. In 2022, the emergence of the Delta variant caused residents to be at higher risk of being infected due to its higher degree of infectivity. There is an urgent requirement to make communities a safe place in which to live.

Safety culture is the product of a group’s values, attitudes and beliefs, competencies, and behavior patterns regarding safety, all of which significantly affect safety performance [1,2]. The current attractiveness of safety culture is linked to the view that safety culture assessment may provide a leading indicator of the level of safety in an organization, potentially making it a possible benchmark for organizational safety performance [3]. At present, people living in rural and urban communities are still at risk of becoming infected with COVID-19 as the epidemic continues to spread around the world. Community-based epidemic prevention and control are related to the safety behavior, safety attitude, and safety awareness of residents, with these constituting the main aspects of organizational safety culture. It is proposed that obtaining a comprehensive understanding of safety culture among residents and taking measures to strengthen the development of safety culture is an effective way to reduce the risk of COVID-19 infection.

Over the past few years, scholars working in safety science have recognized the role of safety culture in shaping safe environments. A significant amount of research has been conducted on the characteristics of safety culture in high-risk organizations. Safety culture was evaluated among employees in high-risk organizations by Zhang et al. [4] on the basis of three coal mines subsidiaries to the Nanyang Mining Company. Cakit et al. [5] assessed the safety culture of five petrochemical production enterprises in Japan. The safety culture among Chinese undergraduates, based on a sample from a university in Beijing, was analyzed by Gong [4]. Monaca et al. [6] conducted an assessment of patients’ safety beliefs, safety values, and safe behaviors, which constitute major aspects of safety culture. Some research has also been performed on safety culture among residents. Desmedt et al. [7] measured the safety culture among residents in six senior living communities in northern Belgium. Similarly, Teigné et al. [8] evaluated the level of safety culture in a French nursing home (also known as senior living community). Lee [9] carried out research on the features of residents’ safety culture in areas surrounding nuclear power plants in Korea. However, research on residents’ safety culture in the context of COVID-19 is limited. One reason for the absence of such research is the lack of an evaluation indicator system and an evaluation method developed for this setting. 

Recently, many methods have been applied for the quantitative assessment of safety culture in high-risk organizations, including statistical analysis [3,10,11], structural equation modeling [5], Bayesian network [12], the comprehensive evaluation method [13,14,15], and other methods. In recent years, the application of comprehensive evaluation methods in safety culture evaluation has increased, because it combines subjective identification and objective laws with regard to certain issues. Accordingly, this study applied a comprehensive evaluation method to evaluate the safety culture of residents in China during the COVID-19 pandemic. The main purpose of this study is to analyze the characteristics of safety culture among residents and the factors constraining the development of safety culture during the COVID-19 pandemic. Similar approaches applied in existing research [13,14,15] have demonstrated positive implementations of comprehensive evaluation methods in the evaluation of safety culture. This highlights the fact that this method can provide a high level of support in the scientific evaluation of safety culture among residents. Such evaluation could be helpful in obtaining a comprehensive understanding of residents’ safety culture and developing more effective strategies for the development of safety culture among residents.

In summary, the purpose of this research is threefold. First, this study aims to establish a set of evaluation indicator systems for safety culture among residents in the context of the COVID-19 pandemic. Second, the primary aim is to assess the safety culture among residents in China during the COVID-19 pandemic using a comprehensive evaluation method. Third, based on the evaluation results, this paper analyzes the characteristics of residents’ safety culture and the factors constraining the development of safety culture among residents. The findings of this study provide an auxiliary decision-making basis in the development of safety culture among residents.

## 2. Materials and Methods

This study aimed to establish an evaluation indicator system for safety culture among residents in the context of the COVID-19 pandemic and to assess safety culture using the comprehensive evaluation method and an obstacle degree model. Based on the results, this paper analyzed the characteristics of safety culture and the factors constraining the development of safety culture among residents. Figure 1 shows the detailed research process, including the following three steps. The first step is the establishment of an evaluation indicator system. Through investigation combined with expert opinions, an evaluation indicator system with three levels was constructed, including a target layer, a criterion layer, and an index layer. The second part is the determination of indicator weight. In this study, the index weight applies the method of combining subjective and objective weights. An analytic hierarchy process is used to calculate the subjective weight, and an entropy weight method is applied for the evaluation of the objective weight to obtain more scientific and fair results. The third step is the process of data preparation. In this paper, a safety culture assessment questionnaire was designed and administered to 300 community residents via the Internet. Based on the results, a comprehensive evaluation and an obstacle diagnosis were conducted for a detailed analysis of the characteristics of safety culture and the factors constraining the development of safety culture among residents in China during the COVID-19 pandemic.

### 2.1. Establishment of Evaluation Indicator System

Through investigation combined with expert opinions, a safety culture evaluation indicator system was constructed from three levels: a target layer, a criterion layer, and an index layer. Among them, the target layer was the safety culture among residents during COVID-19, which was recorded as A; the criterion layer was the four dimensions of organizational safety culture, including institution safety culture, material safety culture, behavior safety culture, and spiritual safety culture, which were recorded as A1~A4, respectively. The analysis of each element is given below. Table 1 shows the complete evaluation system.

Safety culture is a complex system with multiple dimensions [16]. In the existing studies on safety culture dimensions, institution safety culture, material safety culture, behavior safety culture, and spiritual safety culture are included in the safety cultures of various groups [4,11,16,17,18,19]. A report on creating safety cultures in academic institutions indicated that the safety culture dimensions in the academic community were similar to those of other organizations [3]. Therefore, institution safety culture, material safety culture, behavior safety culture, and spiritual safety culture were suggested as the main components of safety culture among community residents in this research, just as in other organizations. Institution safety culture refers to the safety management systems and informal systems of a community (such as safety conventions, folk safety customs, etc.), while material safety culture refers to the materials manufactured and used to ensure safety (such as safety protection tools, equipment, facilities, etc.), as well as safety capital investments. Behavior safety culture refers to safety behaviors, such as social distancing, safety inspections, etc., and spiritual safety culture refers to safety ideology (such as safety perceptions, beliefs, attitudes, etc.), knowledge regarding safety, etc. [20].

Indicators of institution safety culture included the epidemic prevention and safety management in a community. COVID-19 has had a serious impact on community safety and lives. It is important to explore the acceptance of epidemic prevention and control measures in a community [21]. The degree of epidemic prevention and control in a community was thus recorded as one of the evaluation indicators and denoted as A11. Regarding safety management, the “behavior of community managers” indicator represented the efforts managers made in safety management, which was one of the important indicators reflecting the construction of community safety culture. The responsibility of community managers was also recorded as one of the evaluation indicators and denoted as A12. In addition, the safety management system used should also be recorded as one indicator and denoted as A13. Sound safety management systems help to prevent various safety accidents, such as fires and household electricity accidents. The implementation of the safety management system and humanization of the management system used were recorded as A14 and A15, respectively.

Material safety culture mainly included channels for learning safety knowledge, channels for participating in the construction of safety culture, and the status of safety facilities in the community, which were recorded as A21~A23, respectively. Behavior safety culture mainly included the frequency of safety drills, participation in safety education, safety inspections, etc., which were recorded as A31~A36, respectively [14]. Spiritual safety culture mainly includes the safety awareness of residents, the views of residents on safety education, community safety satisfaction of residents, etc., which were recorded as A41~A47, respectively [14].

### 2.2. Determination of Indicator Weight

This study adopted a method of combining subjective and objective weights to obtain more scientific and fair results. The objective weight was evaluated using the entropy weight method, and the subjective weight was calculated by the analytic hierarchy process.

#### 2.2.1. Entropy Weight Method

Figure 2 shows the basic steps of the entropy weight method. First, range transformation was applied for the standardization of the original data. In this study, all indicators were positive indicators. The formula for standardization is given below:(1)$$Y_{ij}=\frac{X_{ij}-X_{min}}{X_{max}-X_{min}}$$

After data standardization, the next step was to calculate the entropy value of the indicator, and the equation was as follows:(2)$$E_{j}=-\frac{1}{\ln m}\overset{m}{\underset{i=1}{\sum }}P_{ij}\ln P_{ij}$$
where m is the total number of evaluation objects, *P_ij_* is the characteristic proportion of the evaluation object under the *j*th indicator, and the characteristic proportion *P_ij_* is calculated by Equation (3):(3)$$P_{ij}=\frac{Y_{ij}}{\sum_{i=1}^{m}Y_{ij}}$$

Finally, the weight coefficient of the *j*th indicator was calculated as follows:(4)$$W_{j}=\frac{1-E_{j}}{k-\sum_{j=1}^{k}E_{j}}$$
where *k* is the number of indicators.

#### 2.2.2. Analytic Hierarchy Process

Figure 3 shows the basic steps of the analytic hierarchy process. First, a clear hierarchy structure was established based on the evaluation indicator system mentioned in Section 2.1. Next, the judgment matrix, *A*, was constructed. Its expression was denoted as follows:(5)$$A=\left[\begin{array}{cccc} a_{11} & a_{12} & \cdots & a_{1n}\\ a_{21} & a_{22} & \cdots & a_{2n}\\ \cdots & \cdots & \cdots & \cdots \\ a_{m1} & a_{m2} & \cdots & a_{mn} \end{array}\right]$$

Third, the maximum characteristic root, *λ_max_*, was solved. Then, consistency tests were completed. The consistency index, *CR*, was calculated as follows:(6)$$CR=\frac{CI}{RI}$$
where *CI* is the consistency index, *RI* is the average random consistency index (Table 2), and the consistency index, *CI*, is calculated by Equation (7):(7)$$CI=\frac{\lambda_{max}}{n-1}$$
where *n* is the order of the matrix, *A*.

In Equation (6), if *CI* < 0.1, the judgment matrix satisfied the consistency requirements, that is the matrix passed the consistency test; otherwise, the judgment matrix will be corrected. Finally, the vector of the maximum characteristic root, *λ_max_*, was solved, which was the weight of the indicator through normalization.

#### 2.2.3. Comprehensive Weighting Method

After calculating the objective and subjective weights, the next step was to calculate the combined weight using Equation (8), and the results were applied in the comprehensive evaluation.
(8)$$u_{j}=\beta z_{j}+\left(1-\beta \right)w_{j}$$

Here, *u_j_* is the combined weight of the *j*th indicator, *z_j_* is the subjective weight, and *w_j_* is the objective weight of the *j*th indicator, and β is the resolution coefficient, usually defined as 0.5 [30,31].

### 2.3. Data Collection and Data Preprocessing

Safety culture in high-risk organizations has traditionally been measured using questionnaire surveys [3]. Additionally, there are some safety culture assessment instruments that have been developed based on previous questionnaires and were tested for effectiveness in their specific research. These previous questionnaires provide good references in terms of both questionnaire structure and question and option settings. Therefore, a residents’ safety culture questionnaire was designed based on the evaluation indicator system established in Section 2.1, combined with previous assessment instruments. The questionnaire consisted of two parts: ① basic information of community residents, including gender, age, education level, etc.; ② community safety culture measurement scale with 21 questions in total. Finally, two questions were also included to screen invalid questionnaires. In addition, all of the participants were encouraged to not answer this question as much as possible.

The survey used a random sample of 300 people from different communities across China. Data collection occurred between Mar. and Jun. 2022. The questionnaire was advertised on social media, with a link invitation or QR code that could be accessed voluntarily, and distributed offline for people not currently using the Internet. A total of 300 participants were told to only fill out in the questionnaire once, and they were informed of the purpose of the study. All fields were marked as mandatory, so a participant could move forward only after answering all the questions, and no data were missing. In addition, all the participants were informed that the collected information would be kept confidential, and the questionnaire was anonymous. No incentives were provided for completing the survey. In addition, because the questionnaire had not been used previously, analysis of reliability and validity was performed using SPSS.

After data collection, the next step was to standardize the evaluation indicators. In the established evaluation indicator system, all the indicators were qualitative indicators. The evaluation indicators should thus be quantified before comprehensive evaluation, including quantification for qualitative characteristics. The subsection interval method was applied for the quantification of qualitative indicators, as shown by Equation (9):(9)$$x_{j}=\{\begin{array}{c} a,A_{i}=Text_{j1}\\ b,A_{i}=Text_{j2}\\ c,A_{i}=Text_{j3}\\ d,A_{i}=Text_{j4}\\ e,A_{i}=Text_{j5} \end{array}$$
where *a*–*e* are constants 0.2, 0.4, 0.6, 0.8, and 1, respectively. *A_i_* is the description of indicators; for example, the responsibility of community managers was divided into five levels.

### 2.4. Comprehensive Evaluation and Obstacle Degree Diagnosis

The comprehensive evaluation model is shown below:(10)$$M_{i}=\sum U_{ij}Y_{ij}$$
where *M_i_* is the level of the safety culture of *i*th evaluation object, *U_ij_* is the combined weight, and *Y_ij_* is the standardized value of the *j*th indicator.

The combined weight of the indicator was calculated using Equation (8). Table 3 shows the evaluation indicator weight of safety culture among residents during COVID-19.

An obstacle factor diagnosis model was also applied to calculate the obstacle degree, which is given in Equation (11) [28].
(11)$$H_{ij}=\frac{I_{ij}F_{j}}{\sum_{j=1}^{m}I_{ij}F_{j}}$$
where *H_ij_* is the obstacle degree of the *j*th indicator in the *i*th evaluation object of the evaluation unit, *I_ij_* is the deviation degree, and *F_j_* is the contribution of the *j*th evaluation indicator to the overall goal. In this study, the weight, *u_j_*, obtained by the above combination weighting method was adopted to represent the contribution of a factor to the overall goal. The deviation degree was calculated using Equation (12):(12)$$I_{ij}=1-Y_{ij}$$

The influence degree of each indicator on the overall evaluation result of residents’ safety culture was then obtained. According to the obstacle degree, the main impact factors affecting safety culture were analyzed.

## 3. Results

In total, 300 valid questionnaires were returned, with a recovery rate of 100%. The reliability was verified using Cronbach’s α, an internal consistency coefficient, and the validity was verified using KMO. In this study, Cronbach’s α coefficient for the 21 items in the questionnaire was 0.897 and the KMO was 0.87, indicating that the reliability met the requirements of the assessment.

### 3.1. Participants’ Characteristics

Figure 4, Figure 5 and Figure 6 show the participants’ characteristics. Ages of participants were categorized under 18 (*n* = 37, 12.33%), from 18 to 25 (*n* = 82, 27.33%), from 26 to 45 (*n* = 85%, 28.33%), from 46 to 60 (*n* = 86, 28.67%), and above 60 (*n* = 10, 3.34%). Level of education included junior high school and below (*n* = 71, 23.67%), senior high (*n* = 62, 20.67%), junior college (*n* = 48, 16%), and bachelor and above (*n* = 119, 39.66%). Distance from infected individuals included residing in the same community (village), the same street (township), the same district (county), the same city, and the same province as infected individuals.

### 3.2. Evaluation of Safety Culture among Residents

The comprehensive evaluation model established in Section 2.4 was used to calculate the level of safety culture among residents. In order to verify the effectiveness of the combined weight method, a comparative analysis between the combined weight method and the AHP method was conducted. Figure 7 shows the results of the model calculation and comparison between the two methods. It can be seen from Figure 7 that similar consistent results were obtained using different models.

The overall level of safety culture among residents was 0.6059, and Figure 8 demonstrates the level of the four dimensions of safety culture among residents. It can be seen from Figure 8 that, during COVID-19, among the four dimensions, the value of institution safety culture was higher than the others. This was because, in the early stage of COVID-19, the epidemic raged out of control, arousing the attention of managers, and they improved the safety management systems of communities. The level of material safety culture was significantly lower than the others.

Figure 9 shows the results of communities with different distances from COVID-19 cases. It can be seen from Figure 9 that the level of safety culture among residents during COVID-19 had an increasing trend with the distance from the infected individuals, mainly because there were differences in risks of virus infection, and the closer the infected individuals, the higher the risk of virus infection.

Figure 10 shows the trends of each dimension with different distances from COVID-19 cases. The results show that, in addition to material safety culture, the values of the remaining three dimensions were strongly related to the distance from infected individuals. For institution safety culture, an explanation for this phenomenon is that the closer the distance, the higher the risk of virus infection and the higher the residents’ requirements for community safety management. Behavior safety culture also had an increasing trend with distance. Similarly, for spiritual safety culture, the results revealed that the level of spiritual safety culture increased with distance.

### 3.3. Obstacle Degree Diagnosis

According to the obstacle degree model, the obstacle degree of each factor to safety culture among residents, as well as among the four dimensions, was calculated using Equation (11), and then the key factors restricting safety culture among residents were analyzed.

Figure 11 shows the obstacle degree of each factor of safety culture among residents, and Figure 12 shows the obstacle degree of each factor for the four dimensions. It can be seen from Figure 11 that the spiritual safety culture was the biggest obstacle restricting safety culture among residents. The main obstacle factors were safety awareness (see Indicator A42) and safety attitudes (see Indicator A41). The above two factors are also the main obstacle factors of safety spiritual culture, as shown in Figure 12. This finding reflects the importance of safety awareness and safety attitude. Second, the result showed that channels for learning knowledge regarding safety were also an obstacle factor (see Indicator A21). This was also the main obstacle factor of material safety culture, as shown in Figure 12. In addition, the implementation of a safety management systems was also an obstacle factor (see Indicator A14). This is also the main obstacle factor of institution safety culture, as shown in Figure 12. The results show the high requirements of residents for community managers. For behavior safety culture, emergency response capacity was an obstacle factor (see Indicator A36), as shown in Figure 12. The results reflected the importance of emergency response capacity, especially during COVID-19.

Figure 13 shows the obstacle degree of each factor to safety culture among residents at different distances. It was found that the obstacle degrees of some factors were strongly related to the distance from infected individuals. For example, epidemic prevention has a decreasing trend with the distance from infected individuals (see Indicator A11), mainly because of the subjective requirements of residents for epidemic prevention. The closer the infected individuals, the higher the risk of virus infection and the higher residents’ requirements for epidemic prevention. Contrarily, safety inspections seems to have an increasing trend (see Indicator A34), mainly because of the different risks of virus infection. The closer the infected individuals, the higher the risk of virus infection, and the stricter the safety inspections. Views on safety education had a decreasing trend with the distance (see Indicator A45), which may also be associated with the risk of virus infection.

It can also be seen from Figure 13 that there were differences in the obstacle factors between communities at different distances. The results show that epidemic prevention, the importance of safety, and community safety satisfaction (Indicators A11, A42, and A47) were the main impact factors of communities with COVID-19 patients. For the communities in the same street (township) or district (county) as infected individuals, safety awareness was the main impact factor, as shown in Figure 13. The main obstacle factors were safety attitudes, safety culture promotion, and knowledge of emergency response for communities with a relatively low risk of virus infection (Indicators A41, A43, and A44).

## 4. Discussion

From the findings of this research, we obtained the characteristics of Chinese residents’ safety culture during COVID-19, and the main obstacle factors of safety culture among residents were found. The results identified the weaknesses of the construction of safety culture and shed light on how to improve safety culture among residents in the Chinese community.

### 4.1. Characteristics of Safety Culture among Residents

In this study, an evaluation indicator system of safety culture among residents was established. Moreover, safety culture among residents was evaluated using the comprehensive evaluation method. The characteristics and its obstacle factors were analyzed. The main finding of this study is that the overall level of safety culture among residents was 0.6059, indicating that safety culture needed to be strengthened. Specifically, in this study, four dimensions were included in the safety culture of residents, including institution safety culture, material safety culture, behavior safety culture, and spiritual safety culture. Among the four dimensions, the value of institution safety culture was significantly higher than the others. The values of material safety culture and spiritual safety culture was lower than others and need to be improved.

Safety education and channels for learning knowledge regarding safety had important effects on safety culture. The above two factors not only had impacts on behavior safety culture and material safety culture (see Indicators A32 and A21), but also affected the construction of spiritual safety culture (see Indicators A41 and A42). This was well illustrated in Gong et al.’s research on safety culture among Chinese undergraduates. They found that students had a good awareness of traffic safety because traffic safety was the main content of the safety education in school, reflecting the effects of safety education and safety promotion on safety awareness and safety attitudes, which were the main aspects of spiritual safety culture [3]. The above results showed that strengthening safety education could have significant effects on maintaining a good spiritual safety culture among residents. Similarly, providing more channels for learning safety knowledge could also help to maintaining a good safety culture. Therefore, managers should make efforts to provide more channels, especially for communities with fewer channels.

The implementation of safety management systems was also an obstacle factor (see Indicator A14). This situation required the managers to first follow a safety management system in a community, and then set a good example for residents. In addition, these results may be due to residents’ lack of the awareness of obeying the safety system. This is consistent with the prior findings of Feroz et al. [17], where some community members had poor compliance with safety measures, and they could inflict damage in their communities. Supervision and reminder may be helpful in improving this situation.

### 4.2. Spatial Differentiation Characteristics of Safety Culture among Residents

The spatial differentiation characteristics of safety culture among residents and their impact factors during COVID-19 in China were also analyzed. This analysis could provide a basis for the construction of safety culture among residents during COVID-19. The above results showed that the level of safety culture was strongly related to the distance from infected individuals, especially for behavior safety culture and spiritual safety culture. This may be because there were differences in the risks of transmission and COVID-19 risk perceptions. This was well illustrated in the research of Chen et al. [32] on the relationship between risk perception of COVID-19 and geographical distance. They found that the risk perceptions of infection were higher in areas closer to the epicenter of COVID-19 in China. Residents who lived in the epicenter of COVID-19 were more concerned about risk of infection.

In terms of behavior, managers and residents responded to the call of epidemic prevention experts to limit their activities and gather less often. The closer to COVID-19 individuals, the fewer the onsite safety education classes, safety drills, and other activities that increased the risk of virus infection. This is well illustrated in research on COVID-19 risk perceptions. People have higher risk perceptions for activities that have been proven to increase the risk of COVID-19 infection, particularly large gatherings and indoor activities [33]. For spiritual safety culture, an explanation for this phenomenon is the poor safety education effect and low participation in safety education. As mentioned earlier, intensive safety education could have significant effects on safety awareness and safety attitude. For institution safety culture, this may be because the closer the distance, the higher the risk of virus infection and the higher the residents’ requirements for community safety management. This was well illustrated in similar research [34]. Residents’ satisfaction with a community was negatively related to their expectations for community management. The correlation between material safety culture and distance was not strong. Since the outbreak of COVID-19, a large amount of workforce and material resources have been invested in community-based epidemic prevention and control. Some problems related to material safety culture construction cannot be solved at this time.

There were also differences in the obstacle factors of safety culture among residents and their impacts on safety culture among residents in different regions. This may be related to the different risks of virus infection. For example, epidemic prevention, the importance of safety, and community safety satisfaction (Indicators A11, A42, and A47, respectively) were the main impact factors of communities with COVID-19 patients. The main obstacle factors were safety attitudes, safety culture promotion, and knowledge of emergency response for communities with relatively low risk of virus infection (Indicators A41, A43, and A44, respectively). On the whole, for communities with a high risk of virus infection, epidemic prevention and safety awareness were the main obstacle factors. However, spiritual safety culture was the main obstacle restricting the safety culture for those at low risk of virus infection. Given the spatial differentiation of residents’ safety culture and the obstacle factors, the following strategies could be formulated.

For communities with COVID-19 patients, the first basic consideration is epidemic prevention. On this basis, attention should be paid to the promotion of online safety education. The priority is epidemic prevention for communities in the same street (township) or district (county) as infected individuals. It is also necessary to supervise and remind residents not to travel to nearby areas with a high risk of virus infection. For communities in the same city or province as the infected individuals, various forms of interesting safety education classes and safety drills should be provided to encourage more residents to participate. It is also vital to make efforts to provide more channels for residents to learn knowledge regarding safety.

### 4.3. Limitations

This study has certain limitations, as follows.

One limitation of this study was the subjectivity of the results for some questions (such as A11 and A12). Specifically, some results were strongly related to the risk of virus infection. For example, epidemic prevention had a decreasing trend with distance from infected individuals (see Indicator A11), mainly because of the subjective requirements of residents for epidemic prevention. This may be caused by unreasonable question and option settings. Some questions’ aims were to evaluate the construction of safety culture from the perspective of residents. When answering such questions, residents may give too much attention to their subjective feelings, which could cause subjective results. Therefore, it may be necessary to revise these questions with relatively strong subjectivity. However, despite declaring this, most of the results reflected the objective situation. The current finding can still reflect the true level of residents’ safety culture during COVID-19 in China and can provide a reference for the construction and improvement of safety culture among residents.

In addition, the safety culture is a complex multi-factor system. Although this study involved the four dimensions of institution, material, behavior, and spiritual safety culture, some factors were not considered. More work needs to be done for the improvement of the evaluation indicator system.

The sample also had some limitations. Because of the severe epidemic situation in China during the survey, the web-based survey was preferred. Considering this, people not currently using the Internet were unavoidably excluded from the sampling process. Therefore, in the sampling process, questionnaires were distributed to people not currently using the Internet by invitations from their families or friends. Participants were encouraged to finish the survey with the help of their families. The offline survey was also conducted to modify the selection bias. The small sample size also leads to the discrepancy between the sample distribution and the actual distribution. According to the data released by the National Bureau of Statistics, the ages of Chinese citizens consist of under 14 (17.95%), 14 to 60 (63.35%), and above 60 (18.7%), while level of education included junior high school and below (65.99%), senior high (16.8%), and junior high school and above (17.22%). It can be seen that the educational levels of most participants were higher. The results of this study showed that the higher the educational level, the higher the level of safety culture. Therefore, the study results may be higher than real practice. This was also illustrated in similar research [17,35] on the safety culture of enterprises. People with different educational backgrounds had different understandings of safety culture. Similarly, people of different ages also had different understandings of safety culture. The safety culture levels of juveniles and the elderly were slightly lower. However, despite declaring this, this research found the relationship between safety culture and geographical distance, as well as the main obstacle factors of safety culture among residents, providing a certain reference value for the future improvement of safety culture.

## 5. Conclusions

This study established an evaluation indicator system of safety culture among residents and conducted a comprehensive evaluation of safety culture among residents for an analysis of the characteristics. An obstacle model was also applied to analyze the main obstacle factors restricting the improvement of safety culture among residents. The following conclusions could be drawn:

(1) The overall level of safety culture among residents was 0.6059, indicating that safety culture needs to be strengthened. The main obstacles affecting safety culture were safety education, channels for learning knowledge regarding safety, and the implementation of safety management systems.

(2) The level of safety culture was strongly related to the distance from infected individuals because of different risks of virus infection. There were also differences in the obstacle factors of residents’ safety culture and their impacts on safety culture for residents in different regions. Given the spatial differentiation of safety culture among residents and its obstacle factors, targeted measures should be implemented to improve safety culture according to the risk of virus infection.

## Figures and Tables

**Figure 1 ijerph-20-01676-f001:**
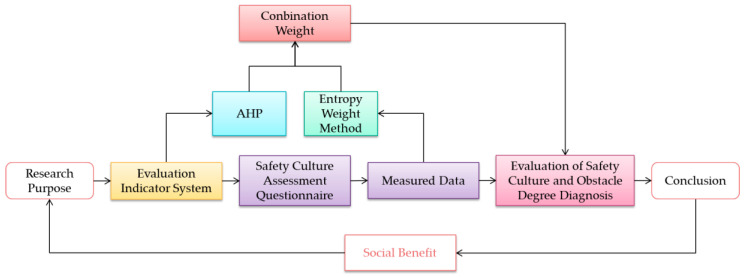
Flowchart of the Research.

**Figure 2 ijerph-20-01676-f002:**
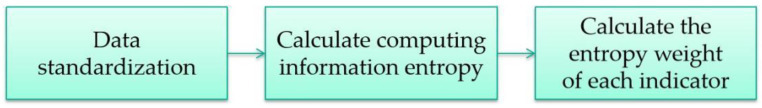
The basic steps of the entropy weight method.

**Figure 3 ijerph-20-01676-f003:**
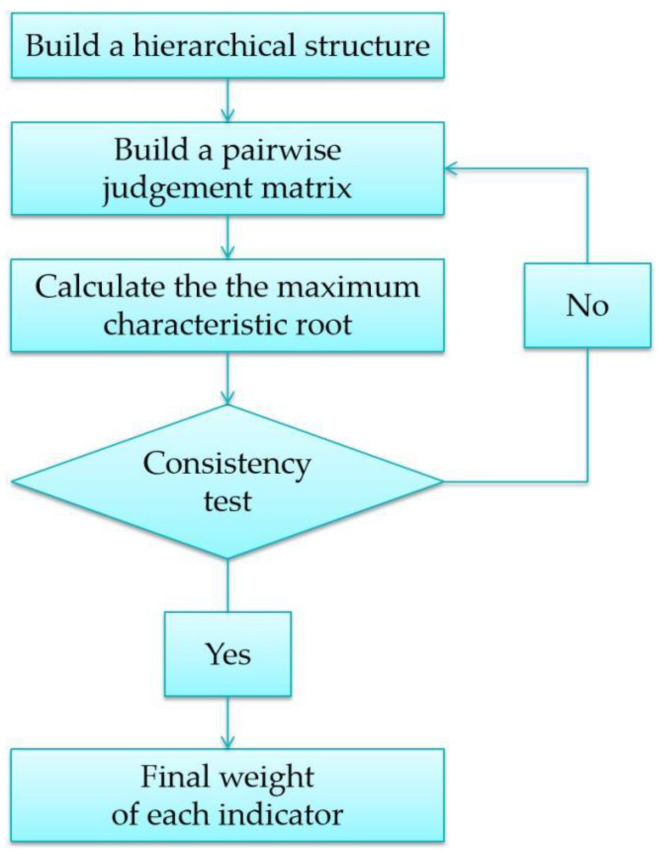
The basic steps of the AHP method.

**Figure 4 ijerph-20-01676-f004:**
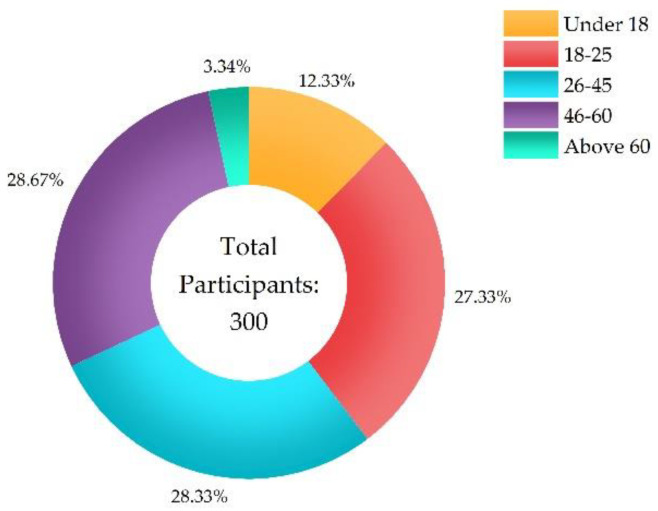
Age of participants.

**Figure 5 ijerph-20-01676-f005:**
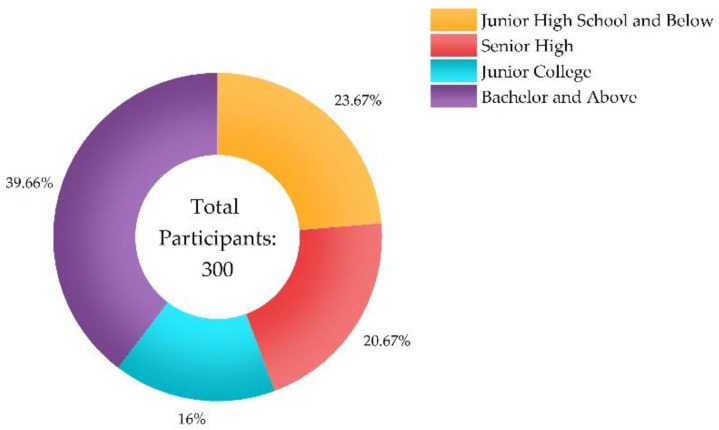
Level of education.

**Figure 6 ijerph-20-01676-f006:**
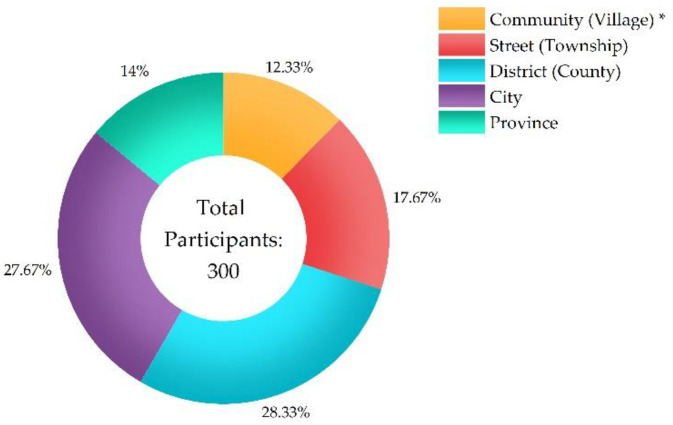
Distance between participants and infected individuals. * “Community (Village)” refers to residing in the same community or village as infected individuals, “Street (Township)” refers to residing in the same street or township as the infected individuals, “District (County)” refers to residing in the same district or county as the infected individuals, “City” refers to residing in the same city as the infected individuals, and “Province” refers to residing in the same province as the infected individuals.

**Figure 7 ijerph-20-01676-f007:**
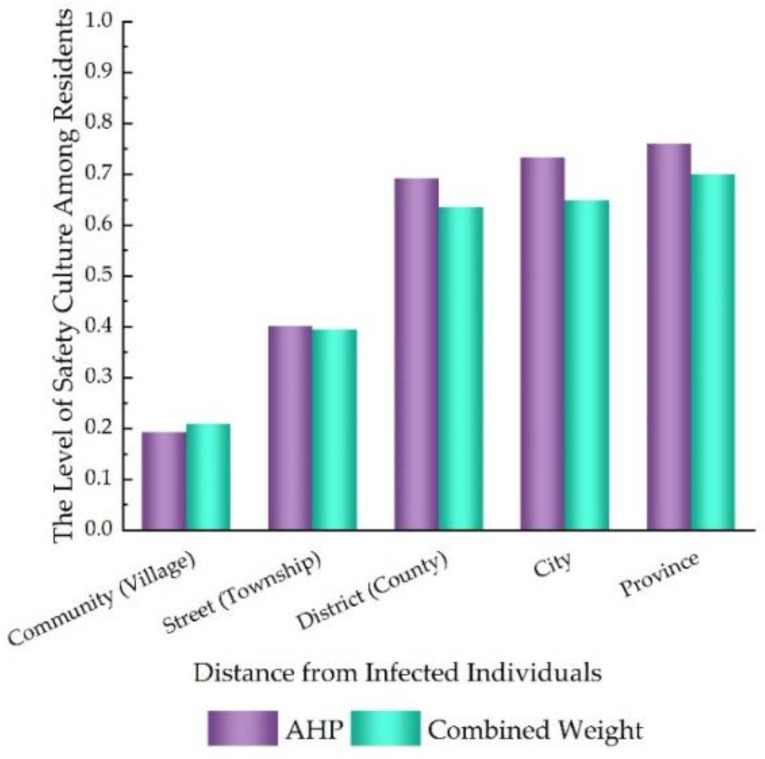
Comparison between different methods.

**Figure 8 ijerph-20-01676-f008:**
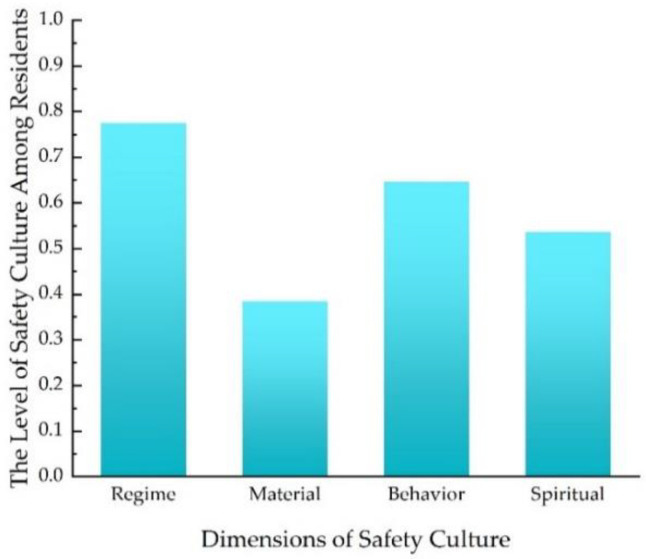
The levels of the four dimensions of culture among residents.

**Figure 9 ijerph-20-01676-f009:**
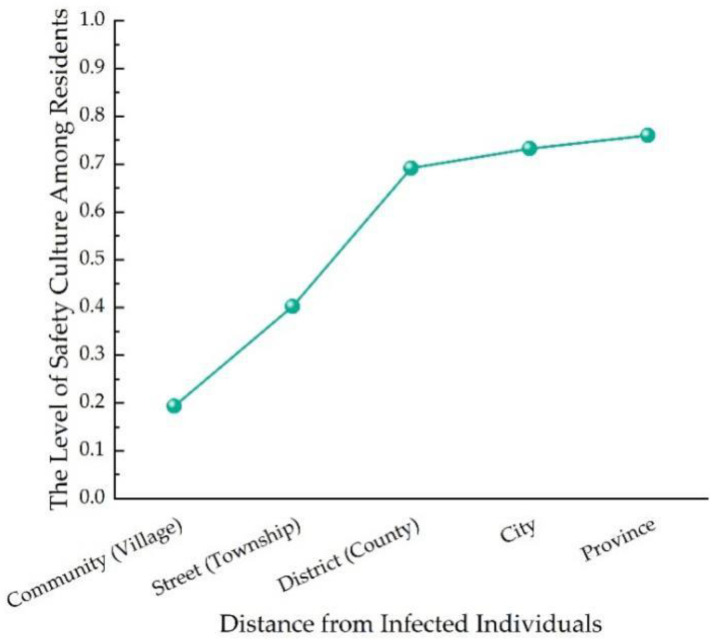
Safety culture of residents in communities with different distances from COVID-19 cases.

**Figure 10 ijerph-20-01676-f010:**
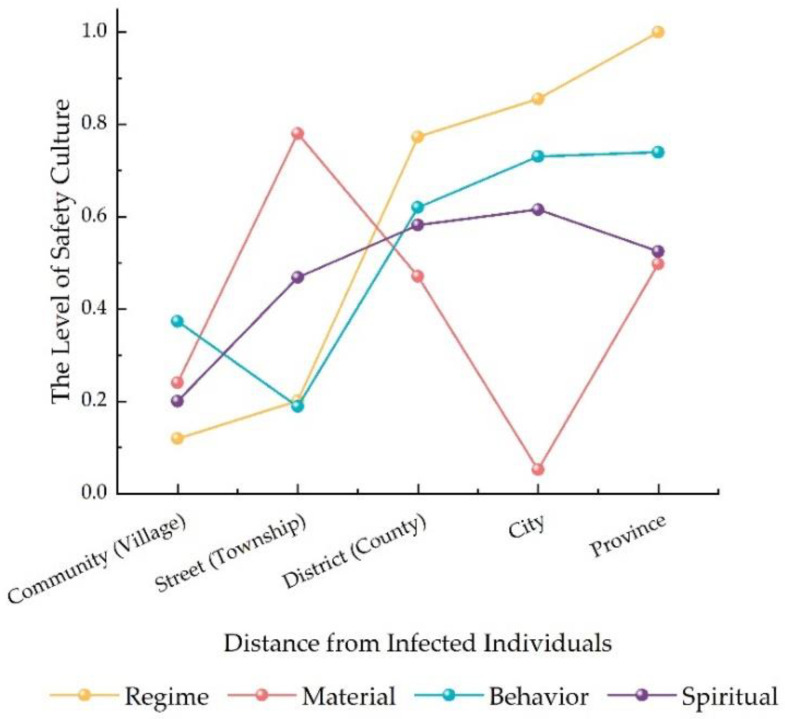
The trends of each dimension with different distance from COVID-19 cases.

**Figure 11 ijerph-20-01676-f011:**
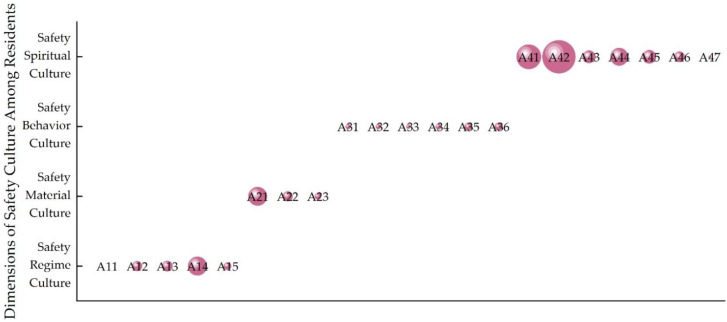
The main obstacle factors of safety culture among residents during COVID-19.

**Figure 12 ijerph-20-01676-f012:**
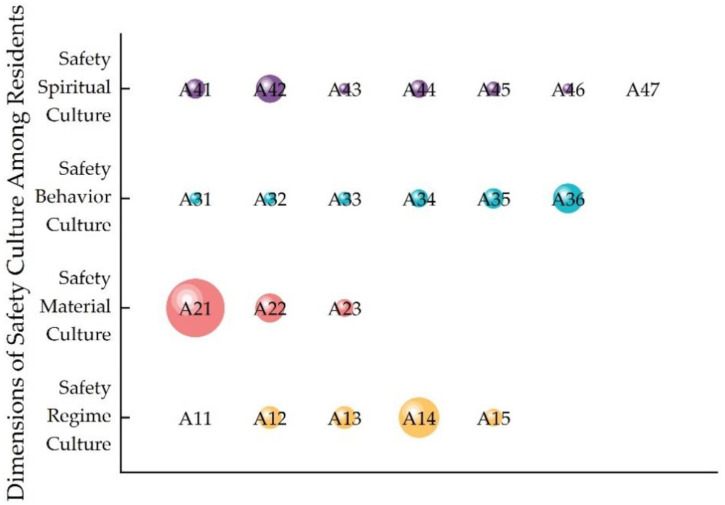
The main obstacle factors of the four dimensions.

**Figure 13 ijerph-20-01676-f013:**
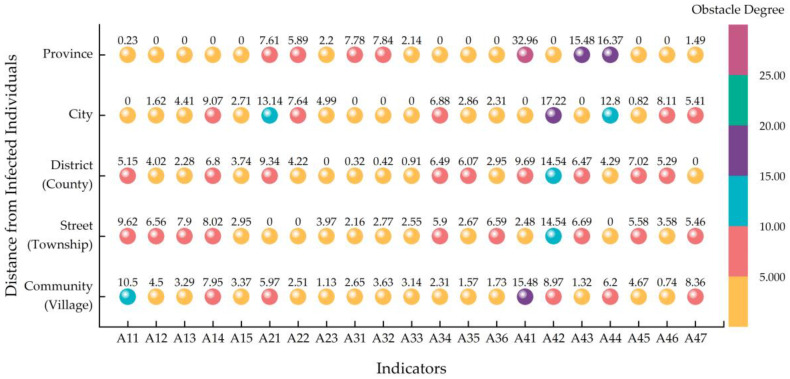
The main obstacle factors of safety culture among residents for different distances.

**Table 1 ijerph-20-01676-t001:** Comprehensive evaluation indicator system of residents’ safety culture during COVID-19.

Target	Primary Indicators	Secondary Indicators	Descriptions	Reference
The Safety culture among residents during COVID-19 A	Safety Institution Culture A1	Epidemic Prevention A11	Degree of epidemic prevention control in a community.	[21]
		Responsibility of Community Manager A12	Responsibility of community managers.	[4,22,23,24]
		Safety Management Systems A13	Whether a safety management system is sound and effective.	[13,19,22,25]
		Implementation of Safety Management Systems A14	Implementation of safety management system.	[4,11,19]
		Humanization of Safety Management Systems A15	Whether a safety management system is humanized.	[4,11,19]
	Safety Material Culture A2	Channels for Learning Knowledge Regarding Safety A21	Channels for learning safety knowledge in a community.	[15,19]
		Channels for Participating in the Construction of Safety Culture A22	Channels to participate in the construction of safety culture in a community.	[26]
		Safety Facilities A23	Status of safety facilities in a community.	[4,11,15,19]
	Safety Behavior Culture A3	Safety Drills A31	Frequency of safety drills.	[21]
		Participation in Safety Education A32	Participation of residents in safety education.	[15,19]
		Effectiveness of Safety Training A33	Effectiveness of safety training.	[4,11]
		Safety Inspections A34	Degree of safety inspections.	[27]
		Feedback on Safety Issues A35	Feedback from managers on safety issues raised by residents.	[26]
		Emergency Response Capacity A36	Emergency response capacity of a community.	[4,11,13]
	Safety Spiritual Culture A4	Importance of Safety A41	Views of residents of safety education on the importance of safety.	[4,11,13,24]
		Safety Mainly Dependent on Safety Awareness A42	Safety awareness of residents.	[4,11,13,24]
		Safety Culture Promotion A43	Promotion of safety culture in a community.	[22]
		Knowledge of Emergency Responses A44	Understanding of emergency response knowledge.	[19]
		Views on Safety Education A45	Views of residents on safety education.	[19]
		Understanding of Safety Identification A46	Understanding of safety identification.	[15]
		Community Safety Satisfaction A47	Community safety satisfaction of residents.	[4,11]

**Table 2 ijerph-20-01676-t002:** Mean random consistency index [28,29].

Matrix Order	1	2	3	4	5	6	7	8
RI	0	0	0.58	0.90	1.12	1.24	1.32	1.41

**Table 3 ijerph-20-01676-t003:** The evaluation indicator weight of safety culture among residents during COVID-19 [28].

Indicator	Objective Weight	Subjective Weight	Combined Weight	Indicator	Objective Weight	Subjective Weight	Combined Weight
A11	0.0398	0.1345	0.0872	A34	0.0455	0.0306	0.0381
A12	0.0508	0.0338	0.0423	A35	0.0405	0.0086	0.0246
A13	0.0333	0.0686	0.051	A36	0.0302	0.0548	0.0425
A14	0.0446	0.0874	0.066	A41	0.0491	0.208	0.1286
A15	0.0394	0.0166	0.028	A42	0.0539	0.1335	0.0937
A21	0.0719	0.0309	0.0514	A43	0.0468	0.0583	0.0526
A22	0.0452	0.0146	0.0299	A44	0.0798	0.0313	0.0556
A23	0.0413	0.0099	0.0256	A45	0.0668	0.0108	0.0388
A31	0.05	0.0028	0.0264	A46	0.0462	0.0173	0.0318
A32	0.0558	0.0045	0.0302	A47	0.0331	0.1058	0.0695
A33	0.0358	0.0163	0.0261				

## Data Availability

Not applicable.

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
