# Peer review of "Research on the Characteristics of Safety Culture and Obstacle Factors among Residents under the Influence of COVID-19 in China"

_ijerph, 2023, doi:10.3390/ijerph20031676_

Round 1

Reviewer 1 Report

It is suggested that the author delete the "suggestion" in the conclusion and suggestion part. The article focuses on discovering scientific laws, and these suggestions can neither be verified nor falsified, so they have little effect.

Reviewer 2 Report

Line 59. The study labelled as "7" and referred as "Melissa et al" doen't fine a correspondence in the bibliography; "Melissa" could be the firest name of the first author of the paper ("Desmedt, M").

Lines 92-93. The meaning of the three adopted "levels" or rather "layers" is not clear in the respect of the definition of the adopted "evaluation indicator system".

Figure 4. No clear definition of the categories "Communiy (Village"), "Street (Township"), "District (Country)", City", "Province".

Lines 94-95. The adopted method for the consultations of the population introduced a clear and relevant selection bias: people not currently using Internet (i.e. people of lower social and cultural class  and / or elderly people) was unavoidably excluded from the sampling process.   

Lines 95-97. The "determination of indicator weight" was the second step to be performed, not the third after the sampling process and the data collecting process.

Figure 1. The flowchart is tortuous and shows several flaws (additional in the respect of the overmentioned selection bias): in particular, the "indicator weight" cell stands at the end of two flowlines coming from "combination weight" and "Evaluation indicator system".

Table 1. No detail about the real meaning of adopted conceptual categories from A11 to A47.

Lines 145 - 155. A total of 300 subjects constitute a very small sample, particularly if intended of representative of "32 provinces and cities and two particular administrative regions" (their population is not indicated). The sentence "Web-based survey options were used to obtain the highest possible response rate" is not clear. The sampling strategy has not been explicated. The refusal rate has not been declared (all the 300 contacted subjects accepted to be interviewed and compiled the questionnaire?).

Figures 6 and 7. The case definition hasn't been declared, nor discussed. The real meaning of the relationship between "the level of safety culture" and the "distance of case" hasn't been exposed, nor discussed: it deals of the hypothesis of a contribution of the knowledge of a diagnosed COVID-19 case inside a community to the "safety culture"? (or rather and better to the "risk perception"?); it deals of the hypothesis of a contribution of the "safety culture" (or rather and better of the "risk perception") to the incidence of COVID-19 infections?     

Reviewer 3 Report

In this paper entitled "Research on the Characteristics in Residents' Safety Culture and Obstacle Factors under the Influence of COVID-19 in China" the authors carry out a comprehensive evaluation indicator model for the residents' safety culture during COVID-19 and the obstacle degree model for the identification of the major factors affecting the residents' safety culture. This work is very interesting and original in an era in which the COVID-19 has depopulated all over the world causing numerous concerns and consequences, therefore I believe that it can have a high interest among readers. Furthermore, the topic is suitable for the IJERPH journal and it is well structured and presented. I have only one suggestion to give to the authors, and for this I accept the paper with minor revisions:

Minor point:

1) The bibliography must be increased, 20 bibliographic references for the length and importance of this paper are few. Authors should add some references, and a single reference at the end of each sentence is not enough.

Round 2

Reviewer 2 Report

A few aspects deserve further improvements (not necessary a further revision after their implementation in the text).

Lines 247-251. The Authors present the age distribution and the levels of education distribution solely for the studied sample. Necessary to give information about the age distribution and the levels of education distribution in the overall population in China too, for the purpose of allowing an evaluation of the practical impact of the selection bias.    

Discussion - section 4.3 "Limitations".

- The selection bias determined by proposing ther questionnaire via Internet has to be discussed (in the Authors' response to the first review report, they  mention some corrective action performed to improve the participation and the compliance of less cultured people and of elderly people, but this issue is not yet reported in the paper).

- The limitations deriving from the very small size of the studied sample (300 subjects), in front of the huge numerousness of Chinese population, have to be discussed.

Two couples of terms have to be disambiguated:

- the Authors sometimes use the term "inspirit" and sometimes use the term "spiritual" presumably for the same concept;

- the Authors sometimes use the term "ruler" and sometimes use the term "regime" presumably for the same concept.
